# FcγRIIB-I232T polymorphic change allosterically suppresses ligand binding

Wei Hu[1†], Yong Zhang[2†], Xiaolin Sun[3†], Tongtong Zhang[1], Liling Xu[4], Hengyi Xie[4], Zhanguo Li[3], Wanli Liu[4*], Jizhong Lou[2,5*], Wei Chen[1,6*]

[1]Department of Neurobiology and Department of Cardiology of the Second Affiliated Hospital, Zhejiang University School of Medicine, Hangzhou, China; [2]Key Laboratory of RNA Biology, CAS Center for Excellence in Biomacromolecules, Institute of Biophysics, Chinese Academy of Sciences, Beijing, China; [3]Beijing Key Laboratory for Rheumatism and Immune Diagnosis (BZ0135), Department of Rheumatology and Immunology, Peking-Tsinghua Center for Life Sciences, Peking University People's Hospital, Beijing, China; [4]MOE Key Laboratory of Protein Sciences, Center for Life Sciences, Collaborative Innovation Center for Diagnosis and Treatment of Infectious Diseases, School of Life Sciences, Beijing Key Lab for Immunological Research on Chronic Diseases, Institute for Immunology, Tsinghua University, Beijing, China; [5]University of Chinese Academy of Sciences, Beijing, China; [6]Key Laboratory for Biomedical Engineering of Ministry of Education, State Key Laboratory for Modern Optical Instrumentation, College of Biomedical Engineering and Instrument Science, Collaborative Innovation Center for Diagnosis and Treatment of Infectious Diseases, Zhejiang University, Hangzhou, China

*For correspondence:
liulab@tsinghua.edu.cn (WL);
jlou@ibp.ac.cn (JL);
jackweichen@zju.edu.cn (WC)

[†]These authors contributed equally to this work

Competing interests: The authors declare that no competing interests exist.

**Abstract** FcγRIIB binding to its ligand suppresses immune cell activation. A single-nucleotide polymorphic (SNP) change, I232T, in the transmembrane (TM) domain of FcγRIIB loses its suppressive function, which is clinically associated with systemic lupus erythematosus (SLE). Previously, we reported that I232T tilts FcγRIIB's TM domain. In this study, combining with molecular dynamics simulations and single-cell FRET assay, we further reveal that such tilting by I232T unexpectedly bends the FcγRIIB's ectodomain toward plasma membrane to allosterically impede FcγRIIB's ligand association. I232T substitution reduces in situ two-dimensional binding affinities and association rates of FcγRIIB to interact with its ligands, IgG1, IgG2 and IgG3 by three to four folds. This allosteric regulation by an SNP provides an intrinsic molecular mechanism for the functional loss of FcγRIIB-I232T in SLE patients.
DOI: https://doi.org/10.7554/eLife.46689.001

## Introduction

Disorders of immune components could lead to autoimmune diseases. Malfunction of an immune receptor, FcγRIIB, is generally destructive for immune system (*Niederer et al., 2010*; *Pincetic et al., 2014*; *Smith and Clatworthy, 2010*). FcγRIIB is widely expressed on most types of immune cells including B cells, plasma cells, monocytes, dendritic cells, macrophages, neutrophils, basophils, mast cells and even memory CD8[+] T cells (*Starbeck-Miller et al., 2014*). Among all the immune-receptors for Fc portion of IgG molecules (FcγRs), FcγRIIB is unique due to its suppressive function against immune cell activation. It has been shown that single-nucleotide polymorphisms (SNPs) of the human FcγRIIB gene extensively influence the susceptibility toward autoimmune disorders (*Kyogoku et al., 2002*; *Niederer et al., 2010*; *Smith and Clatworthy, 2010*). A T-to-C variant in exon 5 (rs1050501) of FcγRIIB causes the I232T substitution (FcγRIIB-I232T) within the transmembrane (TM) domain and

**eLife digest** Left unchecked the immune system can cause devastating damage to healthy tissue. To prevent this from happening, immune cells have built-in off switches that dampen their activation. One such switch is a protein called FcγRIIB that sits on the outer surface of immune cells and binds to proteins known as antibodies, which are produced as part of the immune response. Its role is to act as a brake on the immune system, and stop it from getting out of control.

Overactive immune cells can lead to autoimmune diseases such as systemic lupus erythematosus, also known as SLE for short, which causes damage to the skin, joints and other organs. Previous work suggests that SLE is correlated with a specific mutation in the FcγRIIB gene, but it is unclear how the mutation and the disease are connected.

Proteins are made out of building blocks called amino acids, which have different chemical properties. A swap of one amino acid for another can have big consequences for the structure of a protein. In the case of FcγRIIB, the mutation that correlates with SLE changes an amino acid called isoleucine for another called threonine. Isoleucine does not mix well with water and is commonly found buried in the middle of proteins or inside cell membranes. Threonine, on the other hand, can readily interact with the hydrogen atoms in water and other amino acids.

Hu, Zhang, Sun et al. used computer simulations and imaged single human cells to find out how the isoleucine to threonine change causes immune cells to become over-activated. The experiments revealed that threonine interacts with a nearby amino acid, putting a kink in the FcγRIIB protein. This kink causes the outer part of the FcγRIIB protein to bend towards the immune cell membrane, stopping it from binding to antibodies, and putting a break on immune cells that have become hyper-activated.

There is currently no cure for SLE, but understanding its causes could take us a step closer to better management of the disease. Small molecule drug treatments often target the three-dimensional shape of certain proteins, so understanding the effect of mutations at the molecular level could help with the design of new treatments in the future.

DOI: https://doi.org/10.7554/eLife.46689.002

is positively associated with systemic lupus erythematosus (SLE) in the homozygous FcγRIIB-I232T populations as reported in epidemiological studies (*Chu et al., 2004*; *Clatworthy et al., 2007*; *Kyogoku et al., 2002*; *Niederer et al., 2010*; *Siriboonrit et al., 2003*; *Willcocks et al., 2010*). Although a statistical linkage of the homozygous FcγRIIB-I232T polymorphism with SLE is established, comprehensive assessments and mechanistic investigations towards the inter-linkage of FcγRIIB-I232T regarding to the age of syndrome onset, progress, and clinical manifestation of SLE are still lacking. We first address this question in this report.

Previous biochemical studies revealed that monocytes harboring FcγRIIB-232T (232T) are hyper-activated with augmented FcγRI-triggered phospholipase D activation and calcium signaling (*Floto et al., 2005*). B lymphocytes expressing 232T are of hyperactivity with abnormal elevation of PLCγ2 activation, proliferation and calcium mobilization (*Kono et al., 2005*). 232T-expressing B cells lose the ability to inhibit the oligomerization of B cell receptors (BCRs) upon co-ligation between BCR and FcγRIIB (*Liu et al., 2010*). Recent live-cell imaging studies showed that B cells expressing 232T fail to inhibit the spatial-temporal co-localization of BCR and CD19 within the B cell immunological synapses (*Xu et al., 2014*). Human primary B cells from SLE patients with homozygous FcγRIIB-I232T reveal hyper-activation of PI3K (*Xu et al., 2014*). Thus, it is very likely that FcγRIIB-I232T is the first example that a naturally occurring diseases-associated SNP within the TM domain of a single-pass transmembrane receptor can allosterically suppress the receptor's ligand recognition and signaling functions. FcγRIIB's suppressive function is triggered by its ligand engagement, while this function is disrupted by a single amino acid change in the 232th residue from Ile to Thr in FcγRIIB's TM domain. Two early biochemical studies proposed a model of reduced affinity between 232T and lipid rafts to explain the functional relevance and effect of this natural mutation (*Floto et al., 2005*; *Kono et al., 2005*). A recent study also proposed a different model that I232T mutation enforces the inclination of the TM domain inside the membrane, thereby reducing the lateral mobility and inhibitory functions of FcγRIIB (*Xu et al., 2016*). However, both models assumed

that 232T and FcγRIIB-WT (232I) have an equal capability to perceive and bind to their ligands, the IgG's Fc portion within the antibody-antigen immune complexes. This important but experimentally un-proved pre-requisition in both models is based on the argument that 232T and 232I are identical in the amino acid sequence of their extracellular domains and thus the potential structure of ligand binding site for recognizing the ligands, that is, the IgG's Fc portions (*Dal Porto et al., 2004*; *D'Ambrosio et al., 1996*). However, to date, there is no direct experimental evidence to validate this pre-requisite assumption. We also address this question in this report.

## Results and discussion

In this report, we firstly performed systemic examination over the association of FcγRIIB-I232T with clinical manifestations of SLE. We enrolled 711 unrelated Chinese SLE patients with complete clinical documents into this study (*Supplementary file 1*). 688 unrelated healthy Chinese volunteers with matched gender and age were also enrolled as controls (*Supplementary file 1*). We confirm the presence of a strong positive association of the homozygous FcγRIIB-I232T polymorphism with SLE ($\chi^2$ = 27.224, p=0.008, odds ratio with 95% confidence interval (CI) = 1.927) (*Supplementary file 1*), consistent with the published epidemiological data (*Chu et al., 2004*; *Clatworthy et al., 2007*; *Kyogoku et al., 2002*; *Niederer et al., 2010*; *Siriboonrit et al., 2003*; *Willcocks et al., 2010*). Next, we comprehensively analyzed the clinical data for all 711 SLE patients, including 50 FcγRIIB-I232T homozygotes, 283 FcγRIIB-I232T heterozygotes and 378 FcγRIIB-WT carriers (*Table 1* and *Supplementary file 2*). We find that the homozygous FcγRIIB-I232T polymorphism is significantly associated with early disease onset (age at disease onset <37, p=0.002) (*Table 1*). We also observe a significant association of the homozygous FcγRIIB-I232T polymorphism with more severe SLE clinical manifestations since the corresponding SLE patients present significant elevation in the amounts of anti-dsDNA antibodies (p=0.004), anti-nuclear antibodies (p=0.021) and total Immunoglobulin (Ig) (p=0.032) in comparison to patients carrying heterozygous FcγRIIB-I232T polymorphism or FcγRIIB-WT (*Table 1*). Moreover, homozygous FcγRIIB-I232T polymorphism is also significantly associated with the higher SLE disease activity index (SLEDAI) score (p=0.014 for SLEDAI $\geq$12 vs. p=0.861 for SLEDAI <12) as well as more severe clinical manifestations including arthritis (p=0.008), anemia (p=0.006), leukopenia (p=0.005), complement decrease (p=0.006), hematuria (p=0.004) and leucocyturia (p=0.010) (*Table 1*). A suggestive association is also observed between homozygous FcγRIIB-I232T polymorphism and serositis (p=0.063) (*Table 1*). These clinical association analyses demonstrate that SLE patients homozygous for FcγRIIB-I232T polymorphism are prone to develop more severe clinical manifestations than the patients carrying heterozygous FcγRIIB-I232T polymorphism or FcγRIIB-WT, reinforcing the importance to study the pathogenic mechanism of FcγRIIB-I232T polymorphism since this SNP occurs at a notable frequency in up to 40% (heterozygous polymorphism) humans (*Niederer et al., 2010*; *Smith and Clatworthy, 2010*).

Next, we examined whether I232T polymorphic substitution in the TM domain of FcγRIIB allosterically affects ligand recognition. We did this investigation as our previous observation of the inclination of the TM domain by I232T (*Xu et al., 2016*) led us to hypothesize that tilted TM domain of 232T may lead to ectodomain conformational changes to allosterically attenuate ligand binding. We first carried out large-scale molecular dynamics simulations (MDS) with modeled structures of almost full-length human FcγRIIB (either 232I or 232T) imbedded in the lipid bilayer (*Figure 1A* and *Figure 1—figure supplement 1A*). The simulations confirm our previous results with the MDS of the TM domain of FcγRIIB only (*Xu et al., 2016*), that is, I232T polymorphic substitution enforces the inclination of the TM domain (*Figure 1B*, right). This inclination might result from the ability of H-bond formation between the side-chain Oγ atom of T232 and the backbone oxygen atom of a neighbor residue V228 (*Figure 1B*, left). The difference of the TM domain orientation between 232I and 232T induces a distinct conformation on the ecto-membrane proximal region (ecto-TM linker) (*Figure 1—figure supplement 2*). The membrane buried non-helical region of the linker extends more in 232T than that in 232I. And the length between S218 and P221 peaks at 11 Å for 232T, about 3 Å longer than that for 232I (*Figure 1C*). This length elongation further results in different conformation of residue P217. The main chain dihedral angle of P217 in 232I displays two populations at 141°±23° and −50°±12°, respectively, but shifts to −40°±45° and −75°±12° in 232T (*Figure 1D* and *Figure 1—figure supplement 2*). These effects propagate and lead to a striking effect on tilting FcγRIIB's extracellular domains toward the lipid membrane (*Figure 1E*). Although the extracellular

**Table 1.** Association analysis between homozygous FcγRIIB-I232T genotype and SLE in subphenotype-control cohorts, adjusting for age and sex

| | Genotype frequency | Subphenotype vs. controls | | |
| --- | --- | --- | --- | --- |
| | Tt+tc (%)/CC (%) | OR | 95% | P value |
| Controls | 376+286 (96.2)/26 (3.8) | | | |
| Disease Onset, age < 37 | 207+150 (92.7)/28 (7.3) | 2.739 | 1.456–5.152 | 0.002 |
| Disease Onset, age >= 37 | 171+133 (93.3)/22 (6.7) | 1.657 | 0.872–3.150 | 0.123 |
| Arthritis = 1 | 203+166 (92.5)/30 (7.5) | 2.074 | 1.206–3.565 | 0.008 |
| Arthritis = 0 | 132+87 (94.8)/12 (5.2) | 1.41 | 0.698–2.847 | 0.338 |
| Hematological involvement = 1 | 242+185 (93.4)/30 (6.6) | 1.797 | 1.048–3.084 | 0.033 |
| Hematological involvement = 0 | 90+75 (94.8)/9 (5.2) | 1.369 | 0.627–2.988 | 0.431 |
| Anemia = 1 | 126+94 (91.7)/20 (8.3) | 2.323 | 1.270–4.249 | 0.006 |
| Anemia = 0 | 153+125 (94.2)/17 (5.8) | 1.552 | 0.828–2.907 | 0.17 |
| Leukopenia = 1 | 140+114 (91.7)/23 (8.3) | 2.294 | 1.284–4.099 | 0.005 |
| Leukopenia = 0 | 136+107 (94.9)/13 (5.1) | 1.345 | 0.680–2.663 | 0.394 |
| dsDNA = 1 | 198+151 (92.1)/30 (7.9) | 2.224 | 1.293–3.826 | 0.004 |
| dsDNA = 0 | 129+94 (94.5)/13 (5.5) | 1.48 | 0.747–2.932 | 0.261 |
| ANA = 1 | 308+241 (93.4)/39 (6.6) | 1.82 | 1.094–3.030 | 0.021 |
| ANA = 0 | 25+16 (93.2)/3 (6.8) | 1.893 | 0.549–6.526 | 0.312 |
| Total Ig = 1 | 131+101 (92.8)/18 (7.2) | 1.969 | 1.060–3.66 | 0.032 |
| Total Ig = 0 | 120+88 (93.7)/14 (6.3) | 1.706 | 0.874–3.328 | 0.118 |
| Complement Decrease = 1 | 227+162 (92.4)/32 (7.6) | 2.1 | 1.233–3.578 | 0.006 |
| Complement Decrease = 0 | 58+54 (95.7)/5 (4.3) | 1.124 | 0.423–2.991 | 0.814 |
| Hematuria = 1 | 95+65 (90.9)/16 (9.1) | 2.634 | 1.370–5.063 | 0.004 |
| Hematuria = 0 | 180+140 (94.7)/18 (5.3) | 1.423 | 0.768–2.635 | 0.262 |
| Leucocyturia = 1 | 64+57 (91.0)/12 (9.0) | 2.541 | 1.246–5.178 | 0.010 |
| Leucocyturia = 0 | 196+143 (93.9)/22 (6.1) | 1.651 | 0.922–2.957 | 0.092 |
| Serositis = 1 | 66+45 (92.5)/9 (7.5) | 2.108 | 0.961–4.627 | 0.063 |
| Serositis = 0 | 225+174 (94.1)/25 (5.9) | 1.615 | 0.919–2.838 | 0.096 |
| SLEDAI >= 12 | 20+25 (88.2)/6 (11.8) | 3.327 | 1.273–8.696 | 0.014 |
| SLEDAI < 12 | 116+93 (95.9)/9 (4.1) | 1.072 | 0.493–2.328 | 0.861 |

DOI: https://doi.org/10.7554/eLife.46689.003

domains of 232I and 232T, especially their IgG binding sites, do not undergo obvious conformational change (*Figure 1—figure supplement 3*), their orientations toward the membrane differ significantly. The ectodomain of 232I maintains more straight-up conformation, whereas that of 232T bends down toward the lipid bilayer (*Figure 1E*). Statistical analyses show that the ectodomain inclination angle of 232T distributes across 30 ~ 60° with a sharper single-peak at 40° (*Figure 1E*). In contrast, the angle of 232I distributes much flatter with a favorable probability ranging from 50° to 70° (*Figure 1E*). The distance of C1 domain to the membrane is shorter for 232T than 232I (*Figure 1E*). To check whether these observations result from the thickness of the lipid membrane model, we carried out further simulations using lipids with shorter (14:0/16:1) or longer fatty acid tail (18:0/20:1) (*Figure 1—figure supplement 4A*). The TM helix tilting and S218-P221 prolongation for 232T can be readily observed in these two systems (*Figure 1—figure supplement 4B–4C*). These results suggest that I232T substitution may reduce the ligand recognition ability of FcγRIIB via two aspects. First, the tilting orientation of 232T may sterically prevent the accessibility of the IgG's Fc portion, as significant clashes between docked Fc and the membrane are observed, although FcγRIIB's Fc binding site is not completely buried into the membrane (*Figure 1—figure supplement 1B*). Second, the

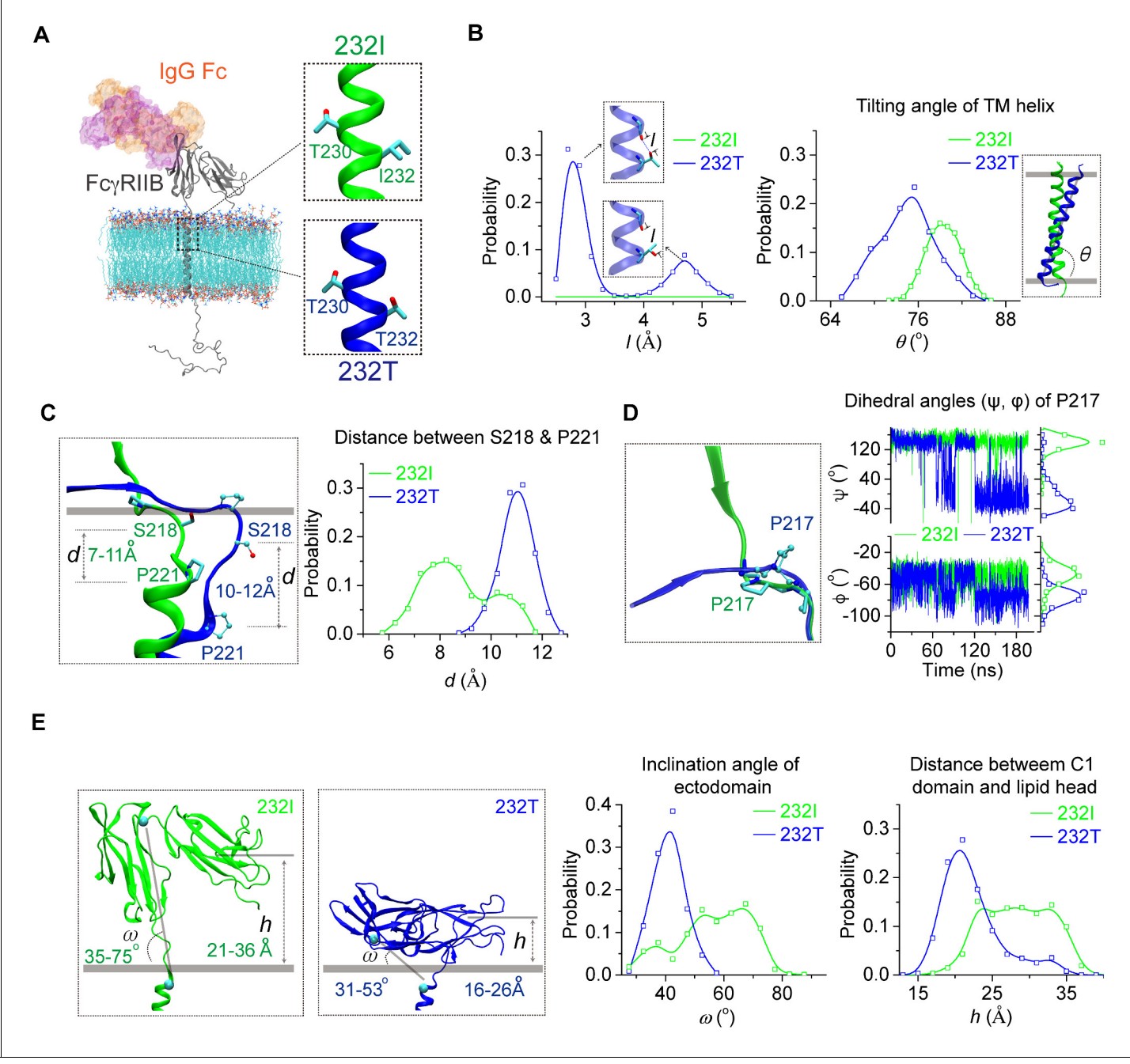

**Figure 1.** MD simulations reveal different conformations of 232I and 232T. (**A**) The modeled structures of almost full-length FcγRIIB (232I and 232T, residues A46-I310, shown in gray cartoon) are complexed with IgG Fc (using the complex structure of IgG Fc and FcγRIIB ectodomain, PDB ID: 3WJJ as a reference, shown in color shaded surface) and imbedded in an asymmetric lipid bilayer (lines with atoms colored by element type: P, tan; O, red; N, blue; C, cyan). The helical structures in the vicinity of residue 232 for 232I (green) and 232T (blue) are shown in the insets. (**B**) Probability distributions of the distance between T232 Oγ atom and its nearest backbone O atom from residue V228 (left), and of the tilting angles between TM helix and lipid bilayer (right). The inclination of TM for 232T can be observed clearly. Blue dashed line in the upper inset of the left panel indicates H-bond. (**C**) The representative snapshot comparison of 232I and 232T at the stalk and TM region by superposing the lipid bilayers (left), and the length distribution of S218-P221 backbone in normal direction of lipid bilayer (right). (**D**) Conformational comparison of I212-S220 regions by aligning residues S218 to S220 (left), and the time courses of the dihedral angles (ψ, φ) of residue P217 (right). (**E**) Representative snapshots of 232I and 232T with the inclination angles and C1(Ig-like C2-Type one domain)/bilayer distances, probability distributions of the inclination angle between FcγRIIB ectodomain and lipid bilayer (left), and the distances between C1 domain and lipid bilayer (right).

DOI: https://doi.org/10.7554/eLife.46689.004

*Figure 1 continued on next page*

*Figure 1 continued*

The following figure supplements are available for figure 1:

**Figure supplement 1.** I232T polymorphic change of FcγRIIB induces the ectodomain recumbent and may impair its binding to Fc portion of antibodies.

DOI: https://doi.org/10.7554/eLife.46689.005

**Figure supplement 2.** Conformation differences of ecto-TM linker and its vicinity between 232I (green) and 232T (blue) are obtained by MD simulations.

DOI: https://doi.org/10.7554/eLife.46689.006

**Figure supplement 3.** IgG-binding sites in both 232I and 232T do not undergo significant structural change.

DOI: https://doi.org/10.7554/eLife.46689.007

**Figure supplement 4.** The length of fatty acid chain and membrane thickness do not alter the effect of I232T polymorphism on the tilting of TM domain.

DOI: https://doi.org/10.7554/eLife.46689.008

ectodomain of 232T is less flexible (*Figure 1E*) such that the chance for FcγRIIB to associate with the ligand is greatly decreased.

We next performed single-cell fluorescence resonance energy transfer (FRET) assay to experimentally validate whether I232T polymorphic change could allosterically bend the FcγRIIB ectodomain toward cell membrane. We fused an mTFP (as FRET donor) at the N-terminal of 232I or 232T ectodomain (mTFP-232I or mTFP-232T) and hypothesized that it should fall in the spatial proximity (~16 ~ 36 Å) for FRET with plasma outer membrane labeled with octadecyl rhodamine B (R18, as FRET acceptor) (*Figure 2A*) and that I232T polymorphism may exhibit an enhanced FRET efficiency. With de-quenching assay (*Chen et al., 2015*; *Xu et al., 2008*) on A20II1.6 B cells expressing similar level of either mTFP-232I or mTFP-232T (*Figure 2B and C*), we find that I232T polymorphic change indeed enhances the FRET efficiency about two folds, from ~20% in 232I to ~40% in 232T (*Figure 2C and D*). This enhancement of FRET efficiency by I232T polymorphism indicates that 232T ectodomain prefers to a more recumbent orientation on the plasma membrane than 232I, consistent with above MDS observations.

Ectodomain orientation change of a receptor can significantly affect its in situ binding affinity with its ligands (*Huang et al., 2004*). We therefore predicted that titling FcγRIIB ectodomain toward plasma membrane by I232T polymorphic change may attenuate its ligand binding affinity, especially the ligand association rate. To test this hypothesis, we applied well-established single-cell biomechanical apparatus with adhesion frequency assay (*Chesla et al., 1998*; *Huang et al., 2010*) to directly and quantitatively measure in situ two-dimensional (2D) binding kinetics of either 232I or 232T binding with its ligands (*Figure 3A*). The results show that the in situ 2D effective binding affinity of 232I with human IgG1 antibody (anti-MERS virus S protein, or anti-S) is about three times higher than that of 232T ($A_cK_a$ = 3.03 ± 0.15×$10^{-7}$ and 0.80 ± 0.04 × $10^{-7}$ μm$^4$, respectively), whereas that with human IgG4 is hardly measured as FcγRIIB and IgG4 binding is known to be extremely weak and beyond the detection limit ($10^{-8}$ μm$^4$) of this assay (*Huang et al., 2010*) (*Figure 3B and F*). Further analyses show that although the 2D off-rates of 232I and 232T from human IgG1 are similar (7.75 ± 1.42 and 7.62 ± 1.41 s$^{-1}$, respectively) (*Figure 3B and H*), the 2D effective on-rate of 232T with IgG1 is three times slower than that of 232I (*Figure 3B and G*). These results are also confirmed by using another human IgG1 antibody (anti-HIV1 gp120 IgG1, or anti-gp120). Both 2D effective affinity and on-rate of 232I with anti-gp120 human IgG1 are three times higher than those of 232T ($A_cK_a$ = 7.74 ± 0.24×$10^{-7}$ and 2.43 ± 0.11 × $10^{-7}$ μm$^4$, respectively; $A_ck_{on}$=5.95±0.19 and 2.16 ± 0.10 × $10^{-7}$ μm$^4$ s$^{-1}$, respectively), while the respective off-rates are similar (7.70 ± 0.83 and 8.90 ± 1.61 s$^{-1}$, respectively) (*Figure 3C and F–H*). I232T polymorphic change also causes the reduction of 2D affinity and on-rate of FcγRIIB's binding with other IgG subtypes (e.g. IgG2 and IgG3) (*Figure 3D–H*).

Furthermore, we aimed to rule out other factors that may potentially reduce ligand binding affinity of 232T. First, we fixed either 232I or 232T expressing B cells to exclude the effect from the reduced lateral diffusion by I232T (*Xu et al., 2016*). We find that fixing B cells expressing FcγRIIB by paraformaldehyde (PFA) does not alter the ligand binding defects for I232T polymorphic change (*Figure 3—figure supplement 1*), suggesting the reduction of lateral diffusion by I232T hardly contributes to the reduction of the ligand binding affinity and on-rate. Moreover, Zhu and colleagues

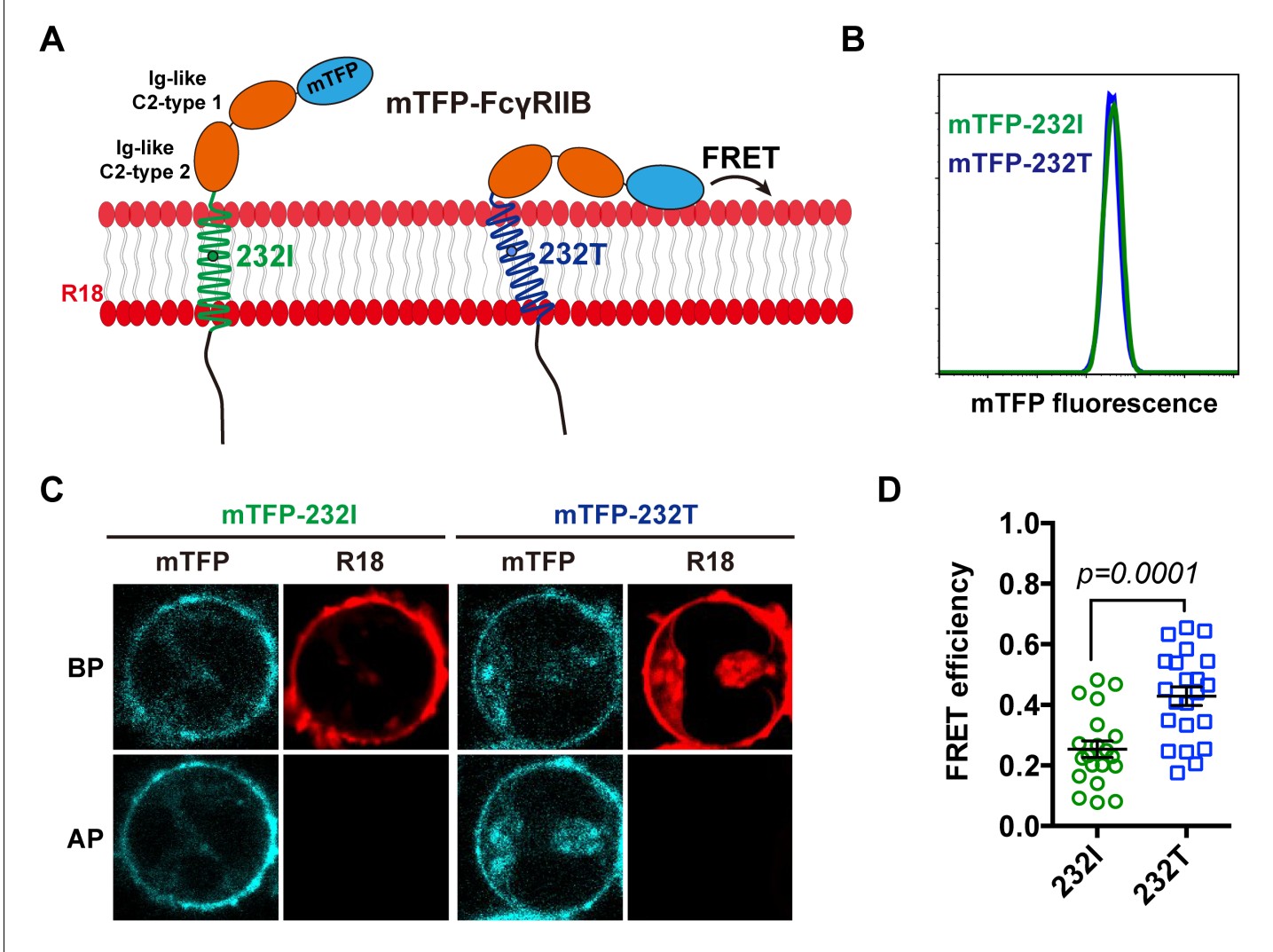

**Figure 2.** The 232T ectodomain prefers to a more recumbent orientation on the plasma membrane. (**A**) Schematic of mTFP-R18 FRET experimental setup to measure the FRET signals between the ectodomain of 232I (green) or 232T (blue) (N-terminal of ectodomain fused with mTFP as FRET donor, cyan) and the plasma membrane (stained with R18 dye as FRET acceptor, red). (**B**) Comparison of mTFP fluorescence intensities of A20II1.6 B cell lines expressing either mTFP-232I (green) or mTFP-232T (blue) constructs by FACS analysis. (**C**) Representative images of de-quenching FRET assay. R18-labeled mTFP-232I or mTFP-232T cell images were acquired in both channels before or after R18 photo-bleaching (BP or AP). (**D**) FRET efficiency comparison of mTFP-232I (green circle) and mTFP-232T (blue square) cells (~20 cells, respectively) with a p-value indicated. Error bars represent SEM.
DOI: https://doi.org/10.7554/eLife.46689.009

have also extensively discussed and experimentally proved that the lateral diffusion has negligible impact on 2D affinity of receptor-ligand binding on live cells (*Chesla et al., 2000*). To further exclude potential technical artifacts in in situ single-cell adhesion frequency assay, we confirm previously known binding-enhancing FcγRIIIA-F158V polymorphism by using the 2D binding assay in this report (*Figure 3—figure supplement 2*). All these data strongly support that I232T polymorphic change in the TM domain of FcγRIIB allosterically tilts FcγRIIB ectodomains toward the plasma membrane, rendering steric hindrance of its ligand binding domain. As a result, 232T exhibits significantly reduced 2D affinity and association on-rate to IgG antibodies. To be noted, it is possible that similar allosteric regulation may be applied to a broad range of transmembrane receptors, for example, potentially explaining how membrane anchor pattern of CD16a influences ligand recognition (*Chesla et al., 2000*).

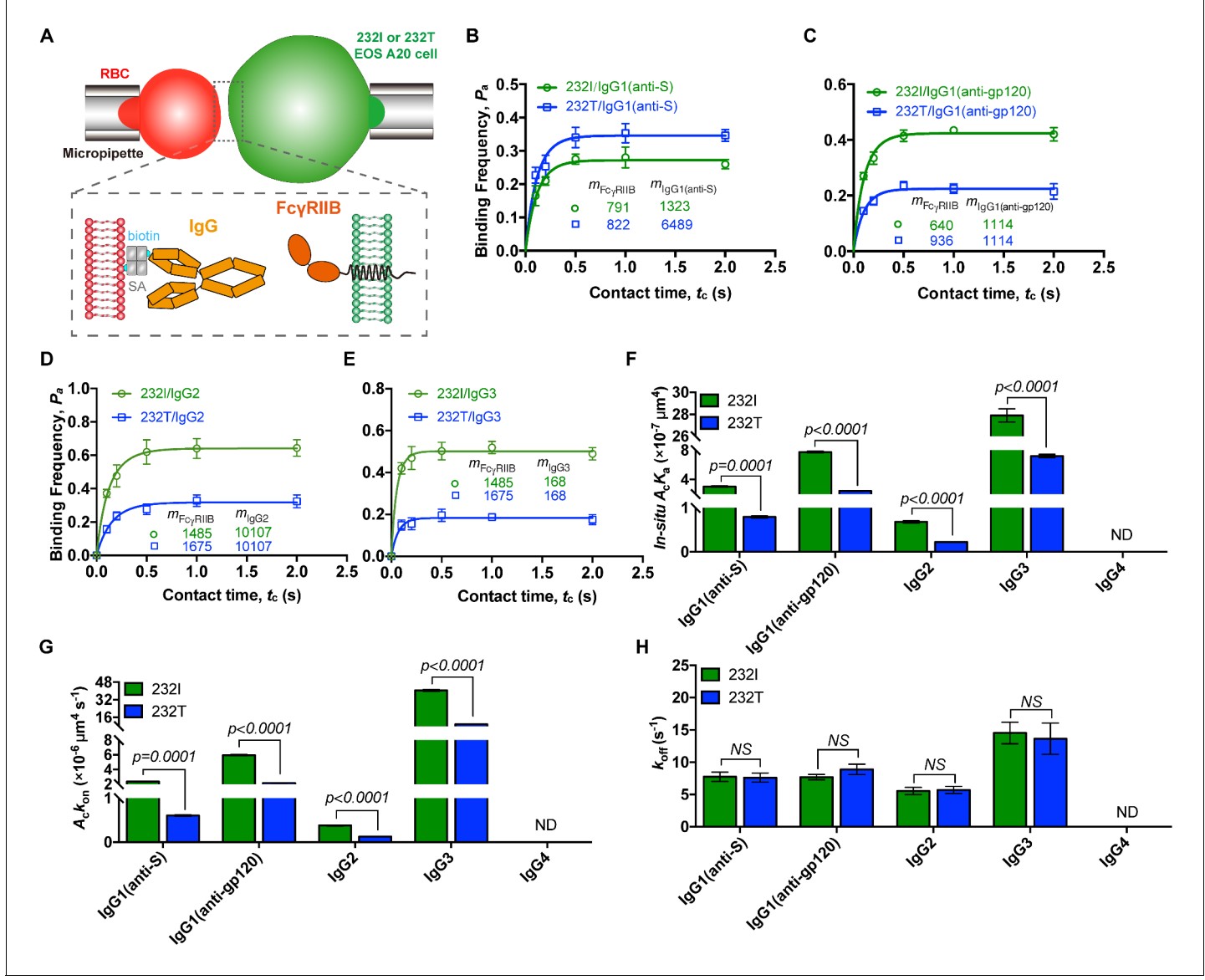

**Figure 3.** 232T exhibits significantly reduced 2D IgG binding affinity and on-rate in comparison with 232I. (**A**) Schematics of experimental setup for single-cell in situ 2D kinetic measurement. Two opposing micropipettes aspirated a human red blood cell coated with a monoclonal IgG (red) and an A20II1.6 B cell expressing either 232I or 232T (green) to operate contact-retraction cycles manipulation, respectively. (**B–H**) Plots of adhesion frequency $P_a$ versus contact time $t_c$ of FcγRIIB (either 232I or 232T) binding with human IgG1 antibody (anti-S, B, or anti-gp120, (**C**)/IgG2 (**D**)/IgG3 (**E**), corresponding in situ 2D effective binding affinity $A_cK_a$ (**F**), on-rate $A_ck_{on}$ (**G**) and off-rate $k_{off}$ (**H**) are compared, respectively. $m_{FcγRIIB}$ and $m_{IgG}$ are surface molecular densities of respective proteins. Error bars represent SEM. ND, not detectable. *NS*, not significantly different.

DOI: https://doi.org/10.7554/eLife.46689.010

The following figure supplements are available for figure 3:

**Figure supplement 1.** 232T exhibits significantly reduced 2D affinity and on-rate of binding to IgG1 (anti-gp120) in comparison with 232I, although FcγRIIB (either 232I or 232T) A20II1.6 B cells are fixed by 4% paraformaldehyde (PFA) plus 4% sucrose.

DOI: https://doi.org/10.7554/eLife.46689.011

**Figure supplement 2.** FcγRIIIA-158V shows a higher 2D affinity than FcγRIIIA-158F in binding with IgG1.

DOI: https://doi.org/10.7554/eLife.46689.012

In summary, we confirm that homozygous FcγRIIB-I232T confers dramatically increased risk of developing more severe clinical manifestations in patients with SLE. The pathological relevant of I232T is caused by the inclination of the TM domain which leads to FcγRIIB ectodomain bending toward plasma membrane, significantly impairing FcγRIIB's binding ability to IgG's Fc portion

through reducing in situ binding affinity and association rate. The hampered Fc recognition ability of FcγRIIB-I232T results in the deficiency on its inhibitory function and thus hyper-activated immune cells, potentially contributing to SLE.

# Materials and methods

**Key resources table**

| Reagent type (species) or resource | Designation | Source or reference | Identifiers | Additional information |
|---|---|---|---|---|
| Cell line (*Mus musculus*) | A20II1.6 B cell | ATCC | | a gift from S.K. Pierce (National Institute of Allergy and Infectious Diseases, Bethesda, MD) |
| Cell line (*Mus musculus*) | 232I A20II1.6 B cell line | (*Xu et al., 2016*) | | |
| Cell line (*Mus musculus*) | 232T A20II1.6 B cell line | (*Xu et al., 2016*) | | |
| Cell line (*Mus musculus*) | mTFP-232I A20II1.6 B cell line | This paper | | Stable mTFP-232I expressing A20II1.6 B cell lines were acquired by lentivirus infection. |
| Cell line (*Mus musculus*) | mTFP-232T A20II1.6 B cell line | This paper | | Stable mTFP-232T expressing A20II1.6 B cell lines were acquired by lentivirus infection. |
| Antibody | Anti-gp120 IgG1, human monoclonal | a kind gift from Dr. Y. Shi, The Institute of Microbiology of the Chinese Academy of Sciences | | Dosage: 20 μg |
| Antibody | Anti-S IgG1, human monoclonal | a kind gift from Dr. L. Zhang and X. Wang, Tsinghua University | | Dosage: 20 μg |
| Antibody | IgG2, human monoclonal | a kind gift from Dr. H. Wang, Hisun, China | RANKL human IgG2 monoclonal antibody (HS629) | Dosage: 20 μg |
| Antibody | IgG3, human monoclonal | InvivoGen | Catalog#bgal-mab3; RRID: AB_2810285 | recombinant Anti-β-Gal-hIgG3 was produced in CHO cells; dosage: 20 μg |
| Antibody | IgG4, human monoclonal | a kind gift from Dr. Y. Shi, The Institute of Microbiology of the Chinese Academy of Sciences | | Dosage: 20 μg |
| Recombinant DNA reagent | PHAGE-mTFP-232I (plasmid) | This paper | | mTFP was fused to N-terminal of FcγRIIB-232I in a pHAGE backbone |
| Recombinant DNA reagent | PHAGE-mTFP-232T (plasmid) | This paper | | mTFP was fused to N-terminal of FcγRIIB-232T in a pHAGE backbone |
| Commercial assay or kit | TIANamp Blood DNA Midi Kit | TIANGEN Biotech, China | Catalog#DP332-01 | TaqMan probe C: 5'-VIC-CGCTACAGCAGTCCCAGT-NFQ-3', TaqMan Probe T: 5'-FAM-CGCTACAGCAATCCCAGT-NFQ-3' |

*Continued on next page*

*Continued*

| Reagent type (species) or resource | Designation | Source or reference | Identifiers | Additional information |
|---|---|---|---|---|
| Commercial assay or kit | TaqMan Genotyping Assays | Life Technology | Catalog#4351376 | |
| Commercial assay or kit | ClonExpress MultiS One Step Cloning Kit | Vazyme, China | Catalog#C113 | |
| Chemical compound, drug | Octadecyl rhodamine B (R18) | Invitrogen | Catalog#O246 | |
| Chemical compound, drug | Biotin-PEG-SGA | JenKem Technology, China | Catalog#ZZ324P050 | |
| Chemical compound, drug | Streptavidin | Sangon Biotech, China | Catalog#C600432 | |
| Chemical compound, drug | EZ-Link Sulfo-NHS-LC-Biotin kits | Thermo Fisher Scientific | Catalog#21435 | |
| Software, algorithm | VMD (Visual Molecular Dynamics) | University of Illinois at Urbana-Champaign | Visual Molecular Dynamics, RRID:SCR_001820 | http://www.ks.uiuc.edu/Research/vmd/ |
| Software, algorithm | NAMD | University of Illinois at Urbana-Champaign | NAMD, RRID:SCR_014894 | http://www.ks.uiuc.edu/Research/namd/ |
| Software, algorithm | CHARMM Force Field | Alex Mackerell lab at School of Pharmacy, University of Maryland | | http://mackerell.umaryland.edu/charmm_ff.shtml |
| Software, algorithm | GraphPad Prism | GraphPad | GraphPad Prism, RRID:SCR_002798 | https://www.graphpad.com/ |

## SNP rs1050501 genotyping and statistical analysis

The ethics committee of Peking University People's Hospital approved this study and informed consents were obtained from each patient and healthy volunteer. All the human-cell-associated experimental guidelines were approved by the Medical Ethics Committee of Peking University People's Hospital (approval no. 2014PHB116-01) and by the Medical Ethics Committee of Tsinghua University (approval no. 20180029). There were 711 patients fulfilling the 1997 revised classification criteria of the American College of Rheumatology that enrolled in this study. Healthy volunteers were recruited as controls. 4–8 ml peripheral blood was acquired from SLE patients and healthy volunteers. Genomic DNA was extracted from peripheral blood samples using the TIANamp Blood DNA Midi Kit (Catalog#DP332-01, TIANGEN Biotech, China) following the manufacturer's protocol. The TaqMan Genotyping Assays were applied for genotyping of SNP rs1050501 (TaqMan probe C: 5'-VIC-CGCTACAGCA GTCCCAGT-NFQ-3', TaqMan Probe T: 5'-FAM- CGCTACAGCA ATCCCAGT-NFQ-3') (Catalog#4351376, Life Technology). Amplification and genotyping analyses were performed using ABI 7300 Real-Time PCR system. Relative quantification of probes levels was calculated (7500 Sequence Detection System Software Version 1.4, ABI). Few samples were genotyped by using primers (forward: 5'-AAGGGGAGCC CTTCCCTCTGTT-3', reverse: 5'-CATCACCCAC CATGTCTCAC-3') binding to the flanking introns of exon five as reported (*Floto et al., 2005*; *Kono et al., 2005*). The DNA sequencing was done by BGI (Beijing). The Pearson chi-square tests were performed for the comparison of differences between cases and controls at genotype model (recessive model CC vs. TT+TC). The odds ratios (OR), 95% confidence intervals (CI) and *p* value for recessive model analysis were calculated using logistic regression, adjusting for age and sex. In statistical analyses, *p* value of less than 0.05 was considered statistically significant.

## Molecular dynamics simulations

Structure models of human FcγRIIB system (residues A46-I310) were built by fusing the crystal structure of the ectodomain (PDB code 2FCB, residues A46-Q215) to the transmembrane (TM) helix (residues M222-R248) model obtained in the previous study (*Xu et al., 2016*). The stalk (residues A216-P221) and cytoplasmic regions (residues K249-I310) were randomly placed. Different initial models were built to minimize possible artifacts in structural modeling. An asymmetric lipid bilayer with the

membrane lateral area of 100 × 100 Å² was generated with Membrane Builder in CHARMM-GUI (*Wu et al., 2014*). The outer leaflet of lipid membrane contained PC, SM, and cholesterol with molar ratio 1:1:1, and the inner leaflet of lipid membrane contained PE, PC, PS, PIP2 and cholesterol with molar ratio 4:3:2:1:5. Different length of the lipid models were used, including the widely used PO series (16:0/18:1) which is the most common lipid within mammalian cell membranes, and other two lipid models with shorter (14:0/16:1) and longer (18:0/20:1) fatty acids. FcγRIIB models were inserted into the lipid membranes with its TM perpendicular to the bilayer surface and the ectodomain stands straight, as shown in *Figure 1A*.

The 232I system was subsequently solvated in 100 × 100 × 203 Å³ rectagular water boxes with TIP3P water model and was neutralized by 0.15 M NaCl. The 232T system was obtained from the same configuration using the Mutator plugin of VMD (*Humphrey et al., 1996*). The final systems contained ~0.20 million atoms in total.

Both systems were first pre-equilibrated with the following three steps: (1) 5000 steps energy minimization with the heavy atoms of proteins and the head group of the lipids fixed, followed by two ns equilibration simulation under one fs timestep with these atoms constrained by five kcal/mol/Å² spring; (2) 5000 steps energy minimization with the heavy atoms of protein fixed, followed by 2 ns equilibration simulation under 1 fs timestep with these atoms constrained by 1 kcal/mol/Å² spring; (3) 4 ns equilibration simulation under 2 fs timestep with the heavy atoms of protein ecto- and TM domains constrained (i.e. the stalk and intracellular portion is free) by 0.2 kcal/mol/Å² spring.

The resulted systems were subjected to productive simulations for 200 ns with 2 fs timestep without any constrains, and the snapshots of the last 80 ns (sampled at 10 ps intervals) were used for detailed analyses including the probability distributions of hydrogen bonds, tilting angles of the TM helix, inclination angles of ectodomain, the distance between Ig-like C2-type one domain and lipid bilayer. The tilting angle of TM helix was defined as the angle between TM helix and membrane plane, similar as that used in previous study (*Xu et al., 2016*). The inclination angle of ectodomain was defined as the angle between the membrane plane and the vector linking N-terminal of TM helix (M222-I224) and linker region of Ig-like C2-type 1 and 2 domain (S130-W132). The distance between Ig-like C2-type one domain and lipid bilayer was defined as the length between center of mass (COM) of this domain and the heavy atoms of phospholipid head in the normal direction of bilayer.

All simulations were performed with NAMD2 software (*Phillips et al., 2005*) using CHARMM36m force field with the CMAP correction (*MacKerell et al., 1998*). The simulations were performed in NPT ensemble (one atm, 310K) using a Langevin thermostat and Nosé-Hoover Langevin piston method (*Feller et al., 1995*), respectively. 12 Å cutoff with 10 to 12 Å smooth switching was used for the calculation of the van der Waals interactions. The electrostatic interactions were computed using the particle mesh Eward method under periodic boundary conditions. The system preparations and illustrations were conducted using VMD.

## Plasmid construction and cell lines establishment

232I and 232T pHAGE plasmids were previously constructed (*Xu et al., 2016*). mTFP was fused to N-terminal of either 232I or 232T in a pHAGE backbone by ClonExpress MultiS One Step Cloning Kit (Catalog#C113, Vazyme, China). Stable mTFP-232I/mTFP-232T expressing A20II1.6 B cell lines were acquired by lentivirus infection (three-vector system: mTFP-232I or mTFP-232I pHAGE, psPAX2, and pMD2.G). A20II1.6 B cell lines expressing similar level of either mTFP-232I or mTFP-232T were obtained by multiple rounds of cell sorting (Beckman moflo Astrios EQ). Either 232I or 232T expressing A20II1.6 B cell line was previously established (*Xu et al., 2016*).

## FRET measurement

FRET measurements were performed as previously described (*Chen et al., 2015*; *Xu et al., 2008*). Briefly, all FRET measurements were carried out on Nikon TiE C2 confocal microscope with 100x oil lens, Argon 457 nm and HeNe 561 nm laser. 1 × 10⁶ mTFP-232I or mTFP-232T expressing A20II1.6 B cells were stained with 300 nM octadecyl rhodamine B (R18) (Catalog#O246, Invitrogen) on ice for 3 min, excited by two lasers sequentially, and imaged before and after R18 photo-bleaching. mTFP intensity was processed through Image J. FRET efficiency was then calculated by (DQ−Q)/DQ,

where DQ and Q are de-quenched and quenched mTFP intensity, respectively. FRET efficiencies of mTFP-232I and mTFP-232T cells (~20 cells, respectively) were calculated and plotted through Prism 7 (GraphPad). Error bars represent SEM.

### RBC preparation

Streptavidin (SA) coated red blood cells (SA-RBCs) preparation have been described previously (*Huang et al., 2010*). Briefly, RBCs freshly collected from finger prick were biotinylated with biotin-PEG-SGA (Catalog#ZZ324P050, JenKem Technology, China) (*Liu et al., 2014*; *Wu et al., 2019*), followed by incubation with streptavidin (Catalog#C600432, Sangon Biotech, China) to make SA-RBCs. Human antibodies (anti-gp120 IgG1 and IgG4, a kind gift from Dr. Y. Shi, The Institute of Microbiology of the Chinese Academy of Sciences; anti-S IgG1, a kind gift from Dr. L. Zhang and X. Wang, Tsinghua University; IgG2, a kind gift from Dr. H. Wang, Hisun, China; IgG3, Catalog#bgal-mab3, InvivoGen) were biotinylated by EZ-Link Sulfo-NHS-LC-Biotin kits (Catalog# 21435, Thermo Fisher Scientific). Different amount of biotinylated IgG was linked into SA-RBCs through SA-biotin interaction at RT for 30 min, respectively to produce IgG-coated RBCs. These IgG-coated RBCs were then used to measure 2D binding kinetics of FcγRIIB/IgG with adhesion frequency assay. All above experimental processes were approved by the institutional ethical review board of Zhejiang University (approval no. 2015–006).

### Adhesion frequency assay

The adhesion frequency assay was applied to measure FcγRIIB/IgG in situ 2D binding kinetics. The detail experimental setup and procedure were previously described (*Huang et al., 2010*). In brief, two opposing micropipettes aspirating the RBC and FcγRIIB-expressing A20II1.6 B cell (either 232I or 232T) respectively were controlled by a customized computer program to operate contact-retraction cycles. Through 50 contact-retraction cycles, the binding frequency $P_a$ was acquired with definite contact area $A_c$ and a series of preset contact time $t_c$ (0.1, 0.2, 0.5, 1, 2 s or longer). 3 ~ 4 cell pairs were tested for each contact time. And these data were then non-linearly fitted to obtain effective 2D binding affinity $A_cK_a$ and off-rate $k_{off}$ by probabilistic kinetic model (*Chesla et al., 1998*):

$$P_a = 1 - exp\{-m_r m_l A_c K_a \left(1 - exp\left(k_{off}\right)\right)\},$$

where $m_r$ and $m_l$ are receptor and ligand densities on cells, respectively. Effective 2D on-rate $A_c k_{on}$ was then calculated as following:

$$A_c k_{on} = A_c K_a \times k_{off}.$$

In order to accurately calculate 2D binding affinity and on-rate, these two molecular densities ($m_{FcγRIIB}$ and $m_{IgG}$) were determined by standard calibration beads (Quantum Alexa Fluor647 MESF kit, Catalog#647, Bangs Laboratories) on flow cytometry (Beckman CytoFLEX S), respectively. Binding kinetics were calculated and plotted through Prism 7 (GraphPad). Error bars represent SEM.

## Acknowledgements

We thank Dr. Y Shi from The Institute of Microbiology of the Chinese Academy of Sciences (IMCAS) for kindly providing us HIV1 gp120 human IgG1 and PD1 human IgG4 antibody, Dr. L Zhang and X Wang from Tsinghua University for kindly providing us MERS virus S protein human IgG1 antibody, Zhejiang Hisun Pharmaceutical Co., LTD and Dr. H Wang for kindly providing us RANKL human IgG2 monoclonal antibody (HS629), core facilities in Zhejiang University School of Medicine for technical supports, especially X Song for FACS supports. This work was supported by grants from the National Basic Research Program of China (2015CB910800 to WChen), the National Science Foundation of China (31470900 and 31522021 to W Chen; 11672317 to J Lou; 11772348 to Y Zhang), the Fundamental Research Funds for the Central Universities (2015XZZX004-32 to W Chen). The computational resources were provided by the National Supercomputing Center Tianjin Center and HPC-Service Station at the Center for Biological Imaging of the Institute of Biophysics.

## Additional information

### Funding

| Funder | Grant reference number | Author |
|---|---|---|
| National Basic Research Program of China | 2015CB910800 | Wei Chen |
| National Natural Science Foundation of China | 31470900 | Wei Chen |
| National Natural Science Foundation of China | 31522021 | Wei Chen |
| National Natural Science Foundation of China | 11672317 | Jizhong Lou |
| National Natural Science Foundation of China | 11772348 | Yong Zhang |
| Fundamental Research Funds for the Central Universities | 2015XZZX004-32 | Wei Chen |

The funders had no role in study design, data collection and interpretation, or the decision to submit the work for publication.

### Author contributions

Wei Hu, Investigation, Methodology, Writing—original draft, Writing—review and editing, Designed the project, Performed FRET and adhesion frequency assay; Yong Zhang, Software, Funding acquisition, Investigation, Methodology, Writing—original draft, Writing—review and editing, Designed the project, Performed MD simulations; Xiaolin Sun, Formal analysis, Investigation, Writing—review and editing, Designed the project, Performed SNP rs1050501 genotyping and statistical analysis; Tongtong Zhang, Investigation, Methodology, Performed FRET and adhesion frequency assay; Liling Xu, Hengyi Xie, Investigation, Methodology, Prepared reagents, Performed antibody biotinylation; Zhanguo Li, Supervision, Methodology, Performed SNP rs1050501 genotyping and statistical analysis; Wanli Liu, Conceptualization, Supervision, Writing—original draft, Project administration, Writing—review and editing, Conceived the project, Designed the project; Jizhong Lou, Conceptualization, Supervision, Funding acquisition, Project administration, Writing—review and editing, Conceived the project, Designed the project, Performed MD simulations; Wei Chen, Conceptualization, Supervision, Funding acquisition, Project administration, Writing—review and editing, Conceived the project, Designed the project

### Author ORCIDs

Yong Zhang (iD) https://orcid.org/0000-0001-6664-435X
Wanli Liu (iD) https://orcid.org/0000-0003-0395-2800
Jizhong Lou (iD) https://orcid.org/0000-0003-4031-6125
Wei Chen (iD) https://orcid.org/0000-0001-5366-7253

### Ethics

Human subjects: The ethics committee of Peking University People's Hospital approved this study and informed consents were obtained from each patient and healthy volunteer. All the human cell associated experimental guidelines were approved by the Medical Ethics Committee of Peking University People's Hospital (approval no. 2014PHB116-01) and by the Medical Ethics Committee of Tsinghua University (approval no. 20180029).

### Decision letter and Author response

Decision letter https://doi.org/10.7554/eLife.46689.017
Author response https://doi.org/10.7554/eLife.46689.018

# Additional files

### Supplementary files
• Supplementary file 1. Association analysis of rs1050501 with SLE (adjusted for sex and age).
DOI: https://doi.org/10.7554/eLife.46689.013
• Supplementary file 2. Demographic characteristic of SLE cohort.
DOI: https://doi.org/10.7554/eLife.46689.014
• Transparent reporting form
DOI: https://doi.org/10.7554/eLife.46689.015

### Data availability
All data generated or analysed during this study are included in the manuscript and supporting files.

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
