## [Decision Letter]

Dear Dr Chen,

Thank you for submitting your article "FcγRIIB I232T polymorphic change allosterically suppresses ligand binding" for consideration by *eLife*. Your article has been reviewed by Tadatsugu Taniguchi as the Senior Editor, a Reviewing Editor, and three reviewers. The following individuals involved in review of your submission have agreed to reveal their identity: Toshiyuki Takai (Reviewer #1); Michael L Dustin (Reviewer #2).

The reviewers have discussed the reviews with one another and the Reviewing Editor has drafted this decision to help you prepare a revised submission.

Summary:

The short report adds to data supporting a connection of I232T polymorphism in FcγRIIB to lupus. The paper then investigates the mechanism by bringing in a new idea from MD simulations that the molecules lies down on the surface and becomes less likely to bind ligand coming from another surface. The authors validate that the N-terminus of the FcR with the I232T is closer to the membrane using a good FRET method, and then use the adhesion frequency assay to determine that the 2D on-rate for initial bond formation is reduced 3-fold with no change in off-rate. This fits very well with the model and may explain other instances where changing the membrane anchor altered 2D kinetic rates.

There are a few issues to clarify.

1) The clinical data is quite different than the biophysical data that follows. It is implied that it adds to prior studies by showing a difference between heterozygous and homozygous for I232T for risk of lupus. Can the authors confirm that this is new finding in relation to the numerous prior studies cited?

2) The modelling of the distortion of the transmembrane-ectodomain interface (Figure 1) is based on a protein section (A216-P221) that is poorly defined structurally. How dependent are the effects on the composition/width of the model membrane? Is this representative of immune cell plasma membranes?

3) Does FcγRIIB-232T affect binding to all IgG subclasses (also IgG2 and IgG3)? One might expect this to be the case if affinity was altered based on differential rigidity/binding site occlusion. If this data is available or could be made available, it would add weight to the findings.

4) The authors might want to cite Xu et al. (2008) if they agree it was first to use this effective FRET pair.

5) That authors might want to cite Chesla et al. (2000) to indicate that others have found difficult to explain results after changing membrane anchor and perhaps this phenomenon might be invested in these situations.

---

## [Author Response]

[…]There are a few issues to clarify.1) The clinical data is quite different than the biophysical data that follows. It is implied that it adds to prior studies by showing a difference between heterozygous and homozygous for I232T for risk of lupus. Can the authors confirm that this is new finding in relation to the numerous prior studies cited?

Thank the reviewers for this comment. *FCGR2B* rs1050501 (c.695T > C) codes an amino acid substitution, Ile232Thr (I232T) on chromosome 1q23.3 (161644048). There are a lot of published epidemiological papers, however those studies just showed that the homozygous genotype for I232T significantly increased the risk of SLE under recessive association model in world-wide populations (Chu et al., 2004; Floto et al., 2005; Kono et al., 2005; Kyogoku et al., 2002; Niederer et al., 2010; Siriboonrit et al., 2003; Willcocks et al., 2010). Although a statistical linkage of the homozygous FcγRIIB-I232T polymorphism with SLE is established, comprehensive assessments investigations towards the correlation of FcγRIIB-I232T regarding to the age of syndrome onset, progress, and clinical manifestation of SLE are still lacking.

In this study, we evaluate the susceptibility of *FCGR2B* rs1050501 to SLE in an enlarged Chinese SLE patient cohort with complete clinical documents. Therefore, for the first time, we establish a significant association of the homozygous FcγRIIB-I232T polymorphism with more severe SLE clinical manifestations in this report. In detail, we provide new information to show that SLE patients carrying homozygous FcγRIIB-I232T present significant elevation in the amounts of anti-dsDNA antibodies (*p*=0.004), anti-nuclear antibodies (*p*=0.021) and total Immunoglobulin (Ig) (*p*=0.032) in comparison to patients carrying heterozygous FcγRIIB-I232T polymorphism or FcγRIIB-WT (Table 1 in the revised manuscript). Moreover, homozygous FcγRIIB-I232T polymorphism is also significantly associated with the higher SLE disease activity index (SLEDAI) score (*p*=0.014 for SLEDAI≥12 vs. *p*=0.861 for SLEDAI<12) as well as more severe clinical manifestations including arthritis (*p*=0.008), anemia (*p*=0.006), leukopenia (*p*=0.005), complement decrease (*p*=0.006), hematuria (*p*=0.004) and leucocyturia (*p*=0.010) (Table 1 in the revised manuscript). A suggestive association is also observed between homozygous FcγRIIB-I232T polymorphism and serositis (*p*=0.063) (Table 1 in the revised manuscript). These clinical association analyses demonstrate that SLE patients homozygous for FcγRIIB-I232T polymorphism are prone to develop more severe clinical manifestations than the patients carrying heterozygous FcγRIIB-I232T polymorphism or FcγRIIB-WT, reinforcing the importance to study the pathogenic mechanism of FcγRIIB-I232T polymorphism since this SNP occurs at a notable frequency in up to 40% (heterozygous polymorphism) humans.

Beyond all these clinical data, in-depth mechanistic investigations towards the inter-linkage of FcγRIIB-I232T regarding to the age of syndrome onset, progress, and clinical manifestation of SLE are still lacking. In the second part of this manuscript, the biophysical study provides a novel mechanism for the pathological relevance of I232T by showing that ectodomain harboring I232T polymorphism bends towards the membrane such that the Fc binding ability is significantly reduced. The hampered Fc recognition ability of FcγRIIB-I232T results in deficiency on its inhibitory function and thus, hyper-activated immune cells, potentially contributing to SLE.

2) The modelling of the distortion of the transmembrane-ectodomain interface (Figure 1) is based on a protein section (A216-P221) that is poorly defined structurally. How dependent are the effects on the composition/width of the model membrane? Is this representative of immune cell plasma membranes?

We fully agree with the reviewer on this point. We did use a structural model for the ectodomain-transmembrane interface, transmembrane and intracellular region of the FcγRIIB in our molecular dynamics simulations (MDS), as a solved structure of the full-length FcγRIIB or this portion alone is not available. To minimize any possible artifact from the modeled structure, we have carried out simulations starting from different initial models, and all these simulations converge into the same results, suggesting the validity and consistency of our simulations. Moreover, in our study, the simulations mainly function to provide testable hypotheses for further validations by all the following biochemical and biophysical experiments. Our main conclusions are drawn corroboratively by both simulations and experimental results.

The reviewer is quite right that the orientation of a transmembrane helix should depend on the composition of the lipid membrane. In our study, the membrane was built as an asymmetric lipid bilayer, with lipid composition mimicking a real plasma membrane of mammalian cells (Marique and Hildebrand, 1973; Pratt et al., 1978; Vanblitterswijk et al., 1982). The head group and the length of fatty acid tails are two key factors for a lipid molecule. In mammalian cells, the lipid head groups in outer leaflet includes mainly PC (phosphocholine) and SM (sphingomyelin), PE (phosphoethanolamine), and other types with minor percentages; the lipid head groups in the inner leaflet commonly includes PE, PC, PS (phosphoserine) and PI (phosphatidylinositol), while some cell membranes also include SM and other rare types of lipid head groups (Ingolfsson et al., 2014; Marrink et al., 2019).

The length of the fatty acid tails vary from 12 to 26 carbons in plasma membrane, with the 16 and 18 carbons tails most abundant (Pratt et al., 1978; Quehenberger et al., 2010; Vanblitterswijk et al., 1982). Cholesterol is also a major component in mammalian cells, and its molar ratio varies in different cell types. Its ratio to phospholipids ranges from 30% to 50 (Marique and Hildebrand, 1973; Pratt et al., 1978; Vanblitterswijk et al., 1982).

The lipid composition we used in our simulations is POPC/PSM/CHOL (1:1:1) for the outer leaflet and POPE/POPC/POPS/POPIP2/CHOL (4:3:2:1:5) for the inner leaflet, which is very close to previous known composition of the cell membrane of mammalian leukocytes (Andoh et al., 2013; Monneron and Dalayer, 1978; Pratt et al., 1978; Vanblitterswijk et al., 1982). The molar ratio of cholesterols is ~33% of total lipids, that is, ~50% for the cholesterol and phospholipids. Thus, our lipid model almost mimics the lipid composition of immune cell plasma membrane. Moreover, in the previous study, using a lipid bilayer model with PC lipids only, which differs significantly with the actual lipid composition in immune cell plasma membrane, the inclination of FcγRIIB-232T (232T) was also observed (Xu et al., 2016). Together with our current study, it can be suggested that our observation of 232T’s TM domain inclination is not affected by the head groups of lipids.

In our original manuscript, we only used 16:18 lipids for fatty acid chain length, which is widely accepted by most MD simulations involving a lipid bilayer. Nevertheless, it is possible that different lengths of the fatty acid chain may result in different thickness of the lipid membrane, which may affect the inclination of the TM domain as pointed out by the reviewer. To address the potential pitfalls, in this revised manuscript, we performed additional simulations with new lipid models with shorter (14:0/16:1) and longer (18:0/20:1) fatty acids respectively (Figure 1—figure supplement 4A, we would like to emphasize here that lipids with long fatty acids tail are very rare in cell membrane as shown in Vanblitterswijk et al., 1982). With these new lipid models, we carried out ~200 ns simulations using the same strategy as that in our initial study for both FcγRIIB-232I (232I) and 232T. The results demonstrate that the thickness of the lipid bilayer is indeed affected by the length of the fatty acid chain (Figure 1—figure supplement 4A), ~3.04 nm for 14:0/16:1 lipids, shorter than the ~3.54 nm for 16:0/18:1 lipids. For 18:0/20:1 lipids, the membrane thickness is further increased to ~4.19 nm (Figure 1—figure supplement 4A). As a result, the shorter lipid tail (14:0/16:1) promotes the most pronounced TM helix tilting (Figure 1—figure supplement 4B), and S218-P221 prolongation (Figure 1—figure supplement 4C). Interestingly, these phenomena can also be observed readily for a bilayer model with longer fatty acid chain (Figure 1—figure supplement 4B,C). These results confirm the validity of original finding.

3) Does FcγRIIB-232T affect binding to all IgG subclasses (also IgG2 and IgG3)? One might expect this to be the case if affinity was altered based on differential rigidity/binding site occlusion. If this data is available or could be made available, it would add weight to the findings.

To further determine whether FcγRIIB-I232T affects its binding to all IgG subclasses (also IgG2 and IgG3), we performed additional single-cell micropipette experiments to measure 2D affinities of 232I or 232T binding with human IgG2 or IgG3 monoclonal antibody respectively (Figure 3).

Our data indeed show that the in situ 2D effective binding affinity of 232I with human IgG2 monoclonal antibody (IgG2) is also about three times higher than that of 232T binding (*A*_c_*K_a_*=6.82 ± 0.58×10^-8^ and 2.26 ± 0.15×10^-8^μm^4^, respectively) (Figure 3F). Moreover, off-rates of 232I and 232T binding with IgG2 are very similar (5.56 ± 1.25 and 5.70 ± 1.24 s^-1^, respectively) (Figure 3H), while the 2D effective on-rate of 232T binding with IgG2 is three times slower than that of 232I (Figure 3G). So, it has similar reduction in both 2D affinity and on-rate by I232T polymorphic change.

This finding is also confirmed by either 232I or 232T binding with human IgG3 monoclonal antibody (IgG3). That is, the 2D effective affinity and on-rate of 232I binding with IgG3 are about four times higher than those of 232T’s binding (*A*_c_*K*_a_=2.79 ± 0.15×10^-6^ and 0.72 ± 0.05×10^-6^μm^4^, respectively; *A*_c_*k*_on_=4.05 ± 0.21 and 0.98 ± 0.07×10^-5^μm^4^s^-1^, respectively), while their binding off-rates are similar (14.53 ± 4.10 and 13.64 ± 5.42 s^-1^, respectively) (Figure 3F-H).

In summary, we confirm that FcγRIIB-I232T polymorphic change indeed reduces the 2D binding affinity and on-rate of FcγRIIB binding to IgG1, IgG2 and IgG3.

4) The authors might want to cite Xu et al. (2008) if they agree it was first to use this effective FRET pair.

We thank the reviewer for pointing this out and agree with their comment. We have updated our references and cited Xu et al. (2008) in the revised manuscript (Results and Discussion section).

5) That authors might want to cite Chesla et al. (2000) to indicate that others have found difficult to explain results after changing membrane anchor and perhaps this phenomenon might be invested in these situations.

We thank the reviewer for this suggestion. Our current finding, as suggested by the reviewer, may provide a mechanistic insight to explain the results reported in this paper. This paper found that membrane anchor pattern (transmembrane or glycosylphosphatidylinositol) influences the two-dimensional kinetic rate of ligand binding (Chesla et al., 2000). It is proposed that the replacement of polypeptide with GPI anchor potentially result in a conformational change of CD16a, although there is no direct evidence to show such conformational change in that paper per se. Here, combining with molecular dynamics simulations and single-cell FRET assay, we provide the first line evidence to directly reveal that the tilting of the TM domain by I232T polymorphic change induces the orientation of the ectodomain of FcγRIIB towards plasma membrane to allosterically impede ligand recognition. As the reviewer suggested, our finding could potentially explain the experimental phenomenon in the original studies by Chesla et al. We have cited this paper and added related discussion into revised manuscript (Results and Discussion section).